# FBXO11 governs macrophage cell death and inflammation in response to bacterial toxins

Yusun Jeon[1], Seong H Chow[1], Isabella Stuart[1], Ashley Weir[2,3], Amy TY Yeung[4], Christine Hale[4,5], Sushmita Sridhar[4,5], Gordon Dougan[4,5], James E Vince[2,3], Thomas Naderer[1]

*Staphylococcus aureus* causes severe infections such as pneumonia and sepsis depending on the pore-forming toxin Panton–Valentine leukocidin (PVL). PVL kills and induces inflammation in macrophages and other myeloid cells by interacting with the human cell surface receptor, complement 5a receptor 1 (C5aR1). C5aR1 expression is tighly regulated and may thus modulate PVL activity, although the mechanisms involved remain incompletely understood. Here, we used a genome-wide CRISPR/Cas9 screen and identified F-box protein 11 (FBXO11), an E3 ubiquitin ligase complex member, to promote PVL toxicity. Genetic deletion of FBXO11 reduced the expression of C5aR1 at the mRNA level, whereas ectopic expression of C5aR1 in FBXO11$^{-/-}$ macrophages, or priming with LPS, restored C5aR1 expression and thereby PVL toxicity. In addition to promoting PVL-mediated killing, FBXO11 dampens secretion of IL-1$\beta$ after NLRP3 activation in response to bacterial toxins by reducing mRNA levels in a BCL-6–dependent and BCL-6–independent manner. Overall, these findings highlight that FBXO11 regulates C5aR1 and IL-1$\beta$ expression and controls macrophage cell death and inflammation following PVL exposure.

## Introduction

The complement system is able to sense and respond to infections leading to the opsonization and removal of invading microbes. Complement activation also results in immune responses that trigger inflammation aimed at persistent pathogens (Reis et al, 2019). In particular, the activation of the complement 5a receptor 1 (C5aR1) by C5a is critical to mount effective immune responses to pathogens as loss of C5aR1 in mice increases susceptibility to some bacterial infections (Calame et al, 2014). Sustained C5aR1 activation, however, contributes to sepsis and sterile inflammatory diseases such as arthritis (Lee et al, 2006; Niederbichler et al, 2006; Rittirsch et al, Li et al, 2017; Herrmann et al, 2018). Thus, human health depends on the tight regulation of C5aR1. This includes post-translational modifications of C5aR1 to control recycling via endosomes and/or degradation within lysosomes (Suvorova et al, 2008). C5aR1 expression is largely restricted to myeloid cells, whereas its detection in other cells remains controversial (Martin, 2007). Besides basal C5aR1 levels in macrophages, its expression can be induced depending on environmental signals. However, comparatively little is known about the factors that regulate C5aR1 induction (Palmer et al, 2012).

Recently, C5aR1 has been identified to promote bacterial infections, as it can act as a receptor for toxins including those secreted by *Staphylococcus aureus* (Spaan et al, 2013, 2014, 2015; Li et al, 2017; Chow et al, 2020). *S. aureus* stably colonizes the skin, noses, and throats of more than 30% of the human population without causing symptoms, but may infect deeper tissues because of catheters, intubation, or immunosuppression (Thwaites et al, 2011; Howden et al, 2023). In particular, methicillin-resistant *S. aureus* strains that express Panton–Valentine leukocidin (PVL) cause severe infections such as necrotizing pneumonia, fasciitis, and sepsis that are increasingly difficult to treat with common antibiotics (Lee et al, 2018). PVL is a pore-forming toxin that kills macrophages and other myeloid cells, thus promoting immune evasion by *S. aureus* (Spaan et al, 2017). The interaction of PVL with human C5aR1 and CD45 triggers complex formation, membrane insertion, and host cell death (Spaan et al, 2013; Tromp et al, 2018). Human C5aR1 determines the cellular specificity of PVL toward myeloid cells and promotes *S. aureus* lung infections in vivo (Chow et al, 2020). Depending on C5aR1, PVL also triggers inflammation in cultured human macrophages and in humanized mice (Holzinger et al, 2012; Chow et al, 2020). This is because PVL activity is sensed by the NLRP3 inflammasome, resulting in pyroptosis, an inflammatory cell death pathway, and the processing and secretion of IL-1$\beta$. Intriguingly, IL-1$\beta$ secretion in macrophages and mice during LPS challenge is affected by C5aR1 signalling, which in turn can be modulated by PVL (Graves et al, 2010; Spaan et al, 2013; Haggadone et al, 2016). Thus, C5aR1 and IL-1$\beta$–mediated inflammation likely contributes to the tissue damage and bacteria dissemination

[1]Department of Biochemistry & Molecular Biology, Monash Biomedicine Discovery Institute, Monash University, Clayton, Australia   [2]The Walter and Eliza Hall Institute of Medical Research, Parkville, Australia   [3]The Department of Medical Biology, University of Melbourne, Parkville, Australia   [4]The Wellcome Sanger Institute, Wellcome Trust Genome Campus, Cambridge, UK   [5]Department of Medicine, University of Cambridge, Cambridge, UK

Correspondence: Thomas.naderer@monash.edu

associated with severe *S. aureus* infections. Despite the critical role of C5aR1 in *S. aureus* infections, it remains unclear whether its expression is modulated and whether this affects PVL activity.

Given that PVL depends on C5aR1 to kill human macrophages, we set out to screen for host factors that regulate C5aR1 expression. By using a genome-wide CRISPR/Cas9 screen in human macrophages treated with PVL, we identified F-box protein 11 (FBXO11). We show here that loss of FBXO11 abrogates C5aR1 expression and thus protects macrophages from PVL killing. Furthermore, we identified that FBXO11 regulates pro-IL-1β levels in a BCL-6–dependent manner. This identifies FBXO11 as a critical regulator of cell death and inflammation and may represent a new thereapeutic target in relevant human diseases.

# Results

## Genome-wide CRISPR/Cas9 screen reveals novel host factors involved in PVL-mediated cell death

To identify other host factors besides C5aR1 that promote cell death in macrophages exposed to PVL, we performed a genome-wide CRISPR screen in PVL-susceptible THP1 macrophages (Fig 1A). After exposure to PVL, surviving macrophages were sorted and analysed by barcoding and sequencing of the target sgRNA region. Functional enrichment analysis of the statistically significant candidate genes (*P*-value < 0.05, Table S1) using DAVID software revealed enrichment of host factors that are involved in various cellular processes including "transcription from RNA polymerase II promoter," "cell adhesion," and "cell–cell signalling" (Fig 1B). Most of the hits were localised to the plasma membrane—including 242 proteins involved in "C-terminus binding" and "integrin binding." The top candidate genes were involved in wide range of biological functions including lipid biosynthesis, gene expression, and post-translational modification (Fig 1B). This included SGMS1 and C5aR1, which have been identified in similar genetic screens previously, thereby validating our screen (Virreira Winter et al, 2016; Tromp et al, 2018).

Here, we focused on a candidate gene that was highly ranked but had not been previously implicated in mediating PVL toxicity and/ or innate immunity—F-box protein 11 (FBXO11). As a member of the F-box protein family, FBXO11 forms part of the E3 ubiquitin ligase Skp, cullin, and F-box (SCF) containing complex, which regulates various cellular processes such as cell cycle regulation, differentiation, and cell death via ubiquitination (Feldman et al, 1997; Skowyra et al, 1997). Ubiquitination is a rapid and reversible post-translational modification that modulates various biological and cellular processes including signal transduction, cell cycle transition, and cellular homeostasis. More recent studies emphasized a role of ubiquitination in regulating innate immunity, with evidence that infectious agents interfere or hijack the host ubiquitination network (Bomberger et al, 2011; Qiu & Luo, 2017).

To validate the screening result, we generated FBXO11 KO THP1 cell lines using independent sgRNAs (Table S2). Two single cell clonal populations were obtained via FACS (namely FBXO11$^{-/-}$ E2 and FBXO11$^{-/-}$ E3) containing deletions in the predicted target sites (Fig 1C). qRT-PCR analysis showed a >2-fold decrease in mRNA levels

in two FBXO11$^{-/-}$ clones compared with WT (Fig 1D). In B cells, FBXO11 exists as two isoforms, isoform 4 and 1, which differ in the N-terminal region. Isoform 4 encodes a longer 927 amino acid protein with a predicted size of 104 kD, whereas isoform 1 encodes 843 amino acid protein and is predicted to be 94 kD. Accordingly, we detected two isoforms in WT THP1 macrophages, which were absent in the two FBXO11$^{-/-}$ clones, as confirmed by Western blot analysis (Fig 1E). The isoforms showed somewhat higher molecular weights in macrophages than observed in other cells, suggesting the presence of additional variants or other mechanisms (Schieber et al, 2020). Loss of FBXO11 did not affect cell survival under normal culture conditions (Fig 1F and G). As expected, both FBXO11$^{-/-}$ clones showed reduced cell cytotoxicity after exposure to PVL when compared with WT (Fig 1G). On the other hand, loss of FBXO11 did not affect susceptibility to another staphylococcal PFT, LukAB, and nigericin, a bacterial toxin which induces rapid pyroptosis–mediated cell death (Fig 1G). This suggests that FBXO11 specifically promotes PVL-mediated cytotoxicity in THP1 macrophages.

## FBXO11 promotes PVL cytotoxicity by regulating cell surface expression of C5aR1

As the loss of FBXO11 affected the susceptibility to PVL but no other toxins, we examined the expression of cell surface receptors targeted by PVL. C5aR1 is required for binding of LukS subunit of PVL (LukS-PV) to the cell surface of macrophages and thereby toxicity (Spaan et al, 2013). Consistent with this, C5aR1$^{-/-}$ THP1 macrophages remained protected from PVL-mediated toxicity (Fig 2A). Flow cytometric analysis revealed that the cell surface expression of C5aR1 in FBXO11$^{-/-}$ was markedly reduced when compared with WT macrophages, although not completely absent as in C5aR1$^{-/-}$ cells (Fig 2B). In contrast, level of another LukS-PV–binding PVL receptor, C5L2 was unaffected (Fig S1). Although levels of CD45 were also decreased in FBXO11$^{-/-}$ macrophages, it remained readily detectable on the cell surface (Fig S1). CD11b remained largely unaffected in FBXO11$^{-/-}$ cells, in agreement with their susceptibility to LukAB (Fig S1). Immunofluorescence analysis of non-permeabilized macrophages indicated heterogenous cell surface expression of C5aR1 in THP1 macrophages which was largely lost in the absence of FBXO11 (Fig 2C). Western blot analysis of total cell lysates likewise revealed a marked reduction, although not complete loss of C5aR1 protein (Fig 2D).

Given the potential role of FBXO11 in controlling protein stability, we next tested whether C5aR1 is susceptible to degradation. Treatment of WT macrophages with the protein synthesis inhibitor, cycloheximide (CHX), showed that C5aR1 levels remained largely unchanged over time and that inhibition of the proteasome with MG132 did not substantially increase C5aR1 levels, suggesting that C5aR1 is not rapidly degraded under normal growth conditions (Fig 2E). Next, we assessed whether the reduced levels of C5aR1 in FBXO11$^{-/-}$ macrophages were apparent at the mRNA level, as an indicator of transcription. qRT-PCR using total RNA extracted from WT and FBXO11$^{-/-}$ macrophages showed that C5aR1 mRNA levels were >2-folds lower in FBXO11$^{-/-}$ than WT control macrophages (Fig 2F). The reduced cell surface expression of C5aR1 in FBXO11$^{-/-}$ macrophages ultimately translated to decreased LukS-PV binding to the cell surface of macrophages and markedly lower toxin levels

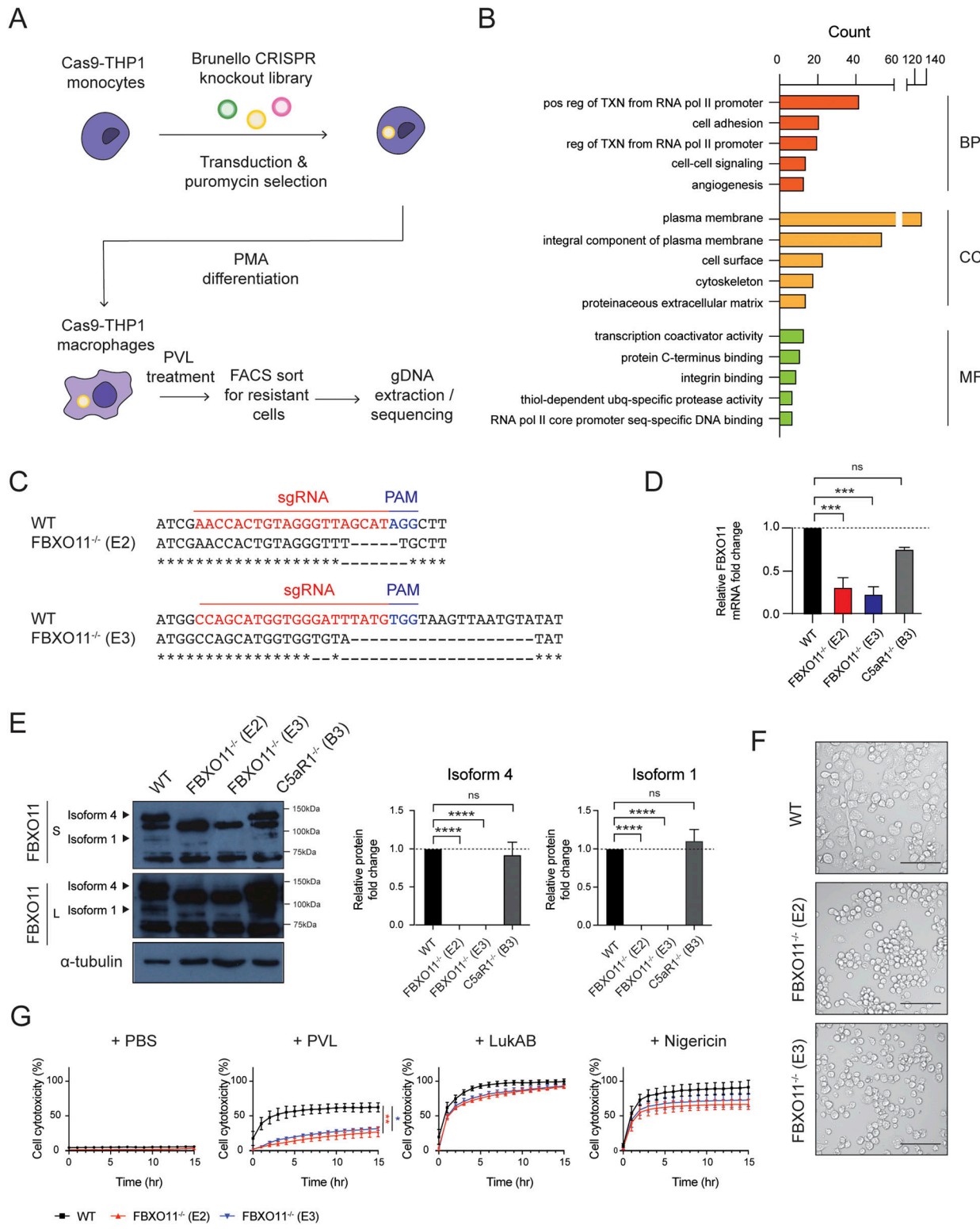

**Figure 1. Genome-wide CRISPR/Cas9 screen identified FBXO11 as critical host factor required for PVL-mediated cell cytotoxicity.**
**(A)** Schematic summary of genome-wide CRISPR/Cas9 screen. **(B)** Top 5 GO term enrichment analysis of 585 statistically significant (*P*-value < 0.05) candidates identified in genome-wide CRISPR/Cas9 screen. Candidates were categorised into biological process (BP), cellular component (CC), and molecular function (MF), and the x-axis represents the gene counts. **(C)** Data from DNA sequencing showing sgRNA target sites (highlighted in red) and PAM sequence (highlighted in blue). **(D)** qRT-PCR analysis of FBXO11 mRNA in WT, FBXO11$^{-/-}$, and C5aR1$^{-/-}$ macrophages. FBXO11 mRNA levels were normalised to GAPDH, and fold change relative to WT shown. Mean ± SEM of three independent biological replicates shown. ns = not significant; *** = *P* < 0.001; by one-way ANOVA followed by Dunnett's multiple comparison test. **(E)** Immunoblot analysis

in total whole cell extracts (Fig 2G and H). Collectively, these results show that reduced susceptibility of FBXO11$^{-/-}$ macrophages to PVL-mediated killing is due to decreased C5aR1 expression.

## C5aR1 expression restores susceptibility of FBXO11$^{-/-}$ macrophages to PVL-mediated death

As a ubiquitin E3 ligase, FBXO11 mediates degradation of its substrates via ubiquitylation and targeting for proteasomal destruction (Duan et al, 2012). FBXO11 must not directly target and ubiquitinate C5aR1 as this would result in increased C5aR1 levels in the absence of the relevant E3 ligase. We thus hypothesized that FBXO11 regulates factors that control the transcription of C5aR1 based on the reduced C5aR1 mRNA levels observed in FBXO11-deficient cells (Fig 2E). To test this idea, we expressed C5aR1 ectopically using doxycycline inducible expression of C5aR1 in FBXO11$^{-/-}$ macrophages (Fig 3A). The doxycycline inducible expression enabled detection of C5aR1 in WT THP1 monocytes, which typically do not express the receptor, and in FBXO11-deficient macrophages, as determined by Western blot analysis of total cell lysates and flow cytometry of cell surface C5aR1 levels (Fig 3B and C). Inducible expression of C5aR1 did not affect CD11b or CD45 expression levels (Fig S2A and B). However, consistent with our hypothesis, C5aR1 expression restored PVL toxicity in FBXO11$^{-/-}$ macrophages (Fig 3D). These data demonstrate that FBXO11 controls the expression of C5aR1 and thus susceptibility to PVL in THP1 macrophages.

## LPS increases C5aR1 expression and restores susceptibility of FBXO11$^{-/-}$ macrophages to PVL-mediated death

Little is known about the transcriptional control of C5aR1, but previous work has indicated that the transcription factor NF-Y may promote its expression after LPS treatment (Hunt et al, 2005). Thus, we next tested whether LPS-mediated up-regulation of C5aR1 expression depends on FBXO11 (Fig 4A). Consistent with previous studies, total C5aR1 protein and mRNA levels were markedly increased in THP1 macrophages after 24 h LPS treatment, but this was largely independent of FBXO11 (Fig 4B and C). In contrast, *S. aureus* and heat-killed *S. aureus* failed to up-regulate C5aR1 in WT and FBXO11$^{-/-}$ macrophages (Fig S3A). LPS-treated FBXO11$^{-/-}$ macrophages restored C5aR1 cell surface expression comparable to WT macrophages (Fig 4D). Consequently, LPS treatment resulted in increased PVL killing of WT and FBXO11-deficient macrophages, although the latter did not reach WT levels (Fig 4E). We thus also monitored the surface expression of CD45. Flow cytometric analysis revealed that LPS treatment did not affect CD45 expression in WT cells and failed to fully restore levels in FBXO11$^{-/-}$ macrophages, potentially explaining the reduced ability of PVL to kill LPS-activated FBXO11-deficient macrophages (Fig S3B). Intriguingly,

LPS caused a significant loss of CD11b cell surface expression, thus protecting WT and FBXO11$^{-/-}$ macrophages from LukAB-mediated killing (Fig S3C and D). Collectively, these findings demonstrate that the transcriptional regulation of C5aR1 in response to LPS regulates PVL toxicity independent of FBXO11.

## FBXO11$^{-/-}$ macrophages show dysregulated IL-1$\beta$ expression

PVL not only causes macrophage cell death but also activates the NLRP3 inflammasome to induce IL-1$\beta$ secretion (Holzinger et al, 2012; Labrousse et al, 2014; Chow et al, 2020). To address whether FBXO11 regulates inflammation after exposure to bacterial toxins, we next determined IL-1$\beta$ secretion as an indicator of NLRP3 activity. In accordance with previous studies, PVL triggered the secretion of IL-1$\beta$ in LPS-primed WT THP1 macrophages, which was slightly reduced in FBXO11$^{-/-}$ macrophages (Fig 5A), as expected based on their decreased expression of its receptor C5aR1 and increased survival rates (Fig 4E). LPS triggered similarly low levels of IL-1$\beta$ secretion in WT and FBXO11$^{-/-}$ macrophages (Fig 5A).

To our surprise, treatment with LukAB or nigericin, a known NLRP3 inflammasome activator, induced higher levels of IL-1$\beta$ release in FBXO11$^{-/-}$ macrophages compared with WT control cells (Fig 5A). IL-1$\beta$ secretion was not significantly affected by loss of C5aR1 (Fig 5B). Given that IL-1$\beta$ is initially expressed as an inactive pro-form (pro-IL-1$\beta$) but is cleaved to the mature form in a caspase-1–dependent manner, we next assessed expression levels of both forms by Western blot analysis. Consistent with the ELISA result, Western blot analysis showed higher levels of mature, cleaved IL-1$\beta$ (17 kD) in the supernatants of LukAB- or nigericin-treated FBXO11$^{-/-}$ macrophages compared with WT macrophages, but not those stimulated with PVL (Fig 5C). We noticed that pro-IL-1$\beta$ (34 kD) was somewhat elevated in the lysates of LPS-primed FBXO11$^{-/-}$ macrophages compared with WT macrophages, although this was not statistically significant (Fig 5C). The difference in pro-IL-1$\beta$ levels, however, was pronounced in unprimed macrophages, whereby FBXO11$^{-/-}$ showed relatively high pro-IL-1$\beta$ levels after differentiation from monocytes, which do not express the cytokine (Fig 5D). The basal NLRP3 level did not markedly differ between WT and FBXO11$^{-/-}$ macrophages, suggesting that the increased IL-1$\beta$ secretion in FBXO11$^{-/-}$ macrophages reflects increased pro-IL-1$\beta$ levels rather than increased NLRP3 inflammasome activity (Fig 5E). We have recently shown that pro-IL-1$\beta$ is rapidly turned over by ubiquitylation and proteasomal targeting in mouse macrophages (Vijayaraj et al, 2021). Consistent with this notion, CHX-mediated inhibition of protein synthesis in WT macrophages resulted in the loss of pro-IL-1$\beta$ levels over time, which was blocked by inhibiting proteasome activity with MG132 (Fig 5F). Pro-IL-1$\beta$ levels in FBXO11$^{-/-}$ macrophages were similarly lower after CHX treatment, although the loss was somewhat delayed in some clones (Fig 5F).

of FBXO11 isoform 4 and 1 in WT, FBXO11$^{-/-}$, and C5aR1$^{-/-}$ macrophages. Protein abundance was normalised to $\alpha$-tubulin and represented as fold change compared with WT macrophages. Mean ± SEM of three independent biological replicates shown. ns = not significant; **** = $P < 0.0001$; by one-way ANOVA followed by Dunnett's multiple Ccomparison test. **(F)** Brightfield microscopy showing morphology of WT and FBXO11$^{-/-}$ macrophages following PMA differentiation. Scale bar corresponds to 100 $\mu$m. **(G)** Live cell imaging showing the percentage of Draq7-positive (dead) WT and FBXO11$^{-/-}$ macrophages treated with PBS, PVL (62.5 ng/ml), LukAB (62.5 ng/ml), and nigericin (10 $\mu$M). Mean ± SEM of three independent biological replicates shown. * = $P < 0.05$ for WT versus FBXO11$^{-/-}$ (E3) at 15 h post toxin treatment; ** = $P < 0.01$ for WT versus FBXO11$^{-/-}$ (E2) at 15 h post toxin treatment; by unpaired $t$ test.
Source data are available for this figure.

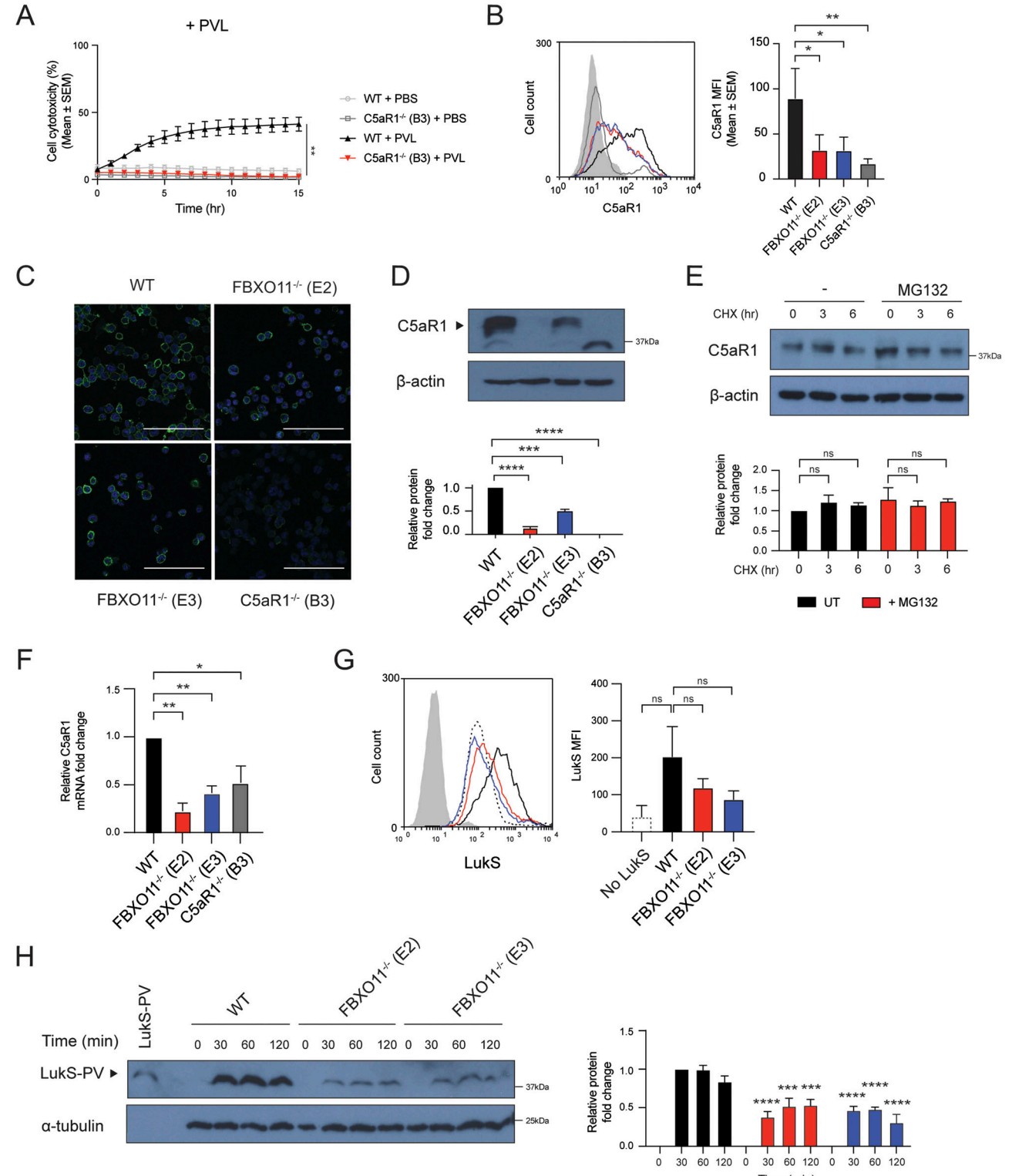

**Figure 2. FBXO11$^{-/-}$ macrophages have reduced C5aR1 cell surface expression.**
**(A)** Live cell imaging showing the percentage of Draq7-positive (dead) WT and C5aR1$^{-/-}$ macrophages to PBS or PVL (62.5 ng/ml). Mean ± SEM of three independent biological replicates shown. ** = $P < 0.01$ for WT versus C5aR1$^{-/-}$ at 15 h post toxin treatment; by unpaired $t$ test. **(B)** Flow cytometric analysis and median fluorescence intensity of C5aR1 cell surface expression. Grey line represents isotype negative control, black line represents WT macrophages, red and blue lines represent FBXO11$^{-/-}$ (E2 and E3, respectively), and dark grey line represents C5aR1$^{-/-}$ macrophages. Mean ± SEM of three independent biological replicates shown. * = $P < 0.05$, ** = $P < 0.01$; by one-way ANOVA followed by Dunnett's multiple comparison test. **(C)** Confocal microscopy images of WT, FBXO11$^{-/-}$, and C5aR1$^{-/-}$ macrophages. PFA-fixed cells were

The similarly short-lived anti-apoptotic factor MCL-1 was degraded in WT and FBXO11$^{-/-}$ macrophages to the same degree (Fig 5F). Tandem ubiquitin binding entity (TUBE) pulldown assays further indicated that FBXO11 was not required for the ubiquitylation of pro-IL-1$\beta$ in both WT and FBXO11$^{-/-}$ macrophages (Fig 5G). We therefore examined IL-1$\beta$ mRNA levels to determine whether FBXO11 regulates pro-IL-1$\beta$ transcription. Indeed, basal IL-1$\beta$ mRNA level in FBXO11$^{-/-}$ macrophages was significantly higher than WT macrophages (Fig 5H). Taken together, these data show that FBXO11 dampens inflammation in macrophages by controlling the transcription of pro-IL-1$\beta$ and thus secretion after exposure to toxins.

### FBXO11 targets BCL-6 for degradation to regulate IL-1$\beta$ levels

Given our findings that FBXO11 affects C5aR1 and IL-1$\beta$ mRNA levels, we next sought to identify targets implicated in regulating their transcription. Among the FBXO11 substrates targeted for proteasomal degradation is BCL-6, a transcriptional repressor/activator critical for B-cell development by preventing CD40 expression (Duan et al, 2012). Consistent with the notion that FBXO11 mediates BCL-6 degradation and thereby promotes CD40 expression (Jiang et al, 2019), FBXO11$^{-/-}$ macrophages contained reduced levels of CD40 compared with WT macrophages, mimicking C5aR1 levels (Fig 6A–C). Next, we used the pharmacological degrader of BCL-6, BI-3802 (Slabicki et al, 2020). As expected, treatment of WT and FBXO11$^{-/-}$ macrophages with BI-3802 resulted in loss of BCL-6 and increased CD40 protein and mRNA levels, although this did not reach statistical significance suggesting other mechanisms are at play (Fig 6D and E). Loss of BCL-6 did not markedly affect C5aR1 levels (Fig 6D and E). In contrast, BI-3802 treatment not only resulted in increased levels of IL-1$\beta$ mRNA and, to some degree, protein in WT but also FBXO11$^{-/-}$ macrophages (Fig 6D and E). Consequently, induced degradation of BCL-6 caused increased IL-1$\beta$ secretion in WT and FBXO11$^{-/-}$ macrophages after exposure to bacterial toxins, including PVL (only in WT), LukAB, and nigericin (Fig 6F). Taken together, these data suggest that BCL-6 regulates IL-1$\beta$ secretion by controlling pro-IL-1$\beta$ mRNA levels.

## Discussion

Recent studies have identified several host cell surface receptors that promote the activity of bacterial pore–forming toxins. For instance, the PVL toxin secreted by *S. aureus* interacts with C5aR1 and CD45, which leads to pore formation, inflammation, and host cell death (Tromp et al, 2018). The identification of these receptors explains the specificity of PVL which primarily kills macrophages, monocytes, and neutrophils, all of which express high levels of C5aR1 and CD45. Although C5aR1 and most other receptors of pore-forming toxins play a core function in immunity, comparatively little is known about how these receptors are regulated, and whether this affects toxin activity. Here, we describe that the E3 ligase FBXO11 promotes the expression of C5aR1 and thus controls the susceptibility of macrophages to PVL. We further demonstrate that FBXO11 regulates the expression of the inflammatory cytokine IL-1$\beta$ and thus NLRP3-mediated inflammation in response to bacterial toxins via BCL-6–dependent and BCL-6–independent manner. CRISPR/Cas9 screens, therefore, not only enable the identification of toxin receptors but also regulatory pathways. As shown here and by others, the expression of toxin receptors within host cells is controlled via multiple mechanisms, involving transcriptional and post-translational steps, which likely affect outcomes in infections (Pacheco et al, 2018; Tian et al, 2018; Yamaji et al, 2019).

*S. aureus* depends on human C5aR1 and other cell surface receptors to establish lung infections and to trigger inflammation (Chow et al, 2020). C5aR1 is mainly expressed on cells belonging to the myeloid lineage. Although human blood monocytes express C5aR1, this is not recapitulated in THP1 or mouse monocytes (Laumonnier et al, 2017). In contrast, several tissue resident human and mouse macrophages express high levels of C5aR1, which was recently confirmed using fluorescent transgenic C5aR1 mice (Karsten et al, 2015). This suggests that C5aR1 expression is controlled by exogenous stimuli and transcriptional programs. The CCAAT-binding factor has been implicated in affecting basal and stimulus-induced (i.e., LPS, IL-4) *C5aR1* gene expression in some but not all macrophages (Burg et al, 1995; Riedemann et al, 2002, 2003; Soruri et al, 2003; Hunt et al, 2005). In contrast, we now demonstrate that C5aR1 expression in THP1 macrophages depends on FBXO11 but not on LPS-activated cells. Based on mRNA abundance and in line with known substrates, we propose here that FBXO11 enables basal *C5aR1* transcription in THP1 macrophages. FBXO11 has been shown to transcriptionally control other proteins via neddylation or direct targeting of transcription factors such as p53, BCL-6, and CDT-2 for degradation (Baron et al, 1993; Abida et al, 2007; Abbas & Dutta, 2011; Rossi et al, 2013; Tateossian et al, 2015). It is frequently mutated or deleted in lymphomas of germinal centre origins leading to

stained using anti-C5aR1 (green), and nuclei were stained with DAPI (blue). Scale bar corresponds to 100 $\mu$M. **(D)** Immunoblot analysis of C5aR1 in WT, FBXO11$^{-/-}$, and C5aR1$^{-/-}$ macrophages. Protein abundance was normalised to $\beta$-actin and represented as fold change compared with WT macrophages. Mean ± SEM of three independent biological replicates shown. *** = $P < 0.001$, **** = $P < 0.0001$; by one-way ANOVA followed by Dunnett's multiple comparison test. **(E)** Immunoblot analysis of C5aR1 in WT macrophages that were treated with CHX (20 $\mu$g/ml) with or without MG132 for the indicated amount of time before cell lysate collection. Protein abundance was normalised to $\beta$-actin and represented as fold change compared with WT macrophage. Mean ± SEM of three independent biological replicates shown. ns = not significant; by two-way ANOVA with Sidak's multiple comparisons test. **(F)** qRT-PCR analysis of C5aR1 mRNA in WT, FBXO11$^{-/-}$, and C5aR1$^{-/-}$ macrophages. C5aR1 levels were normalised to GAPDH, and fold change relative to WT macrophage shown. Mean ± SEM of three independent biological replicates shown. * = $P < 0.05$, ** = $P < 0.01$; by one-way ANOVA followed by Dunnett's multiple comparison test. **(G)** Flow cytometric analysis and MFI of recombinant LukS-PV subunit binding to WT and FBXO11$^{-/-}$ macrophages. Grey line represents isotype, dotted black line represents no LukS-PV negative control, solid black line represents WT macrophages, and red and blue lines represent FBXO11$^{-/-}$ macrophages treated with LukS-PV (E2 and E3, respectively). *** = $P < 0.001$, **** = $P < 0.0001$; by two-way ANOVA with Sidak's multiple comparisons test. **(H)** Western blot analysis of WT and FBXO11$^{-/-}$ macrophages treated with LukS-PV over time. Protein abundance was normalised to $\alpha$-tubulin and represented as fold change compared with WT macrophages at respective time points. Recombinant LukS-PV is included in the left lane. Mean ± SEM of three independent biological replicates shown. *** = $P < 0.001$, **** = $P < 0.0001$; by two-way ANOVA with Sidak's multiple comparisons test.
Source data are available for this figure.

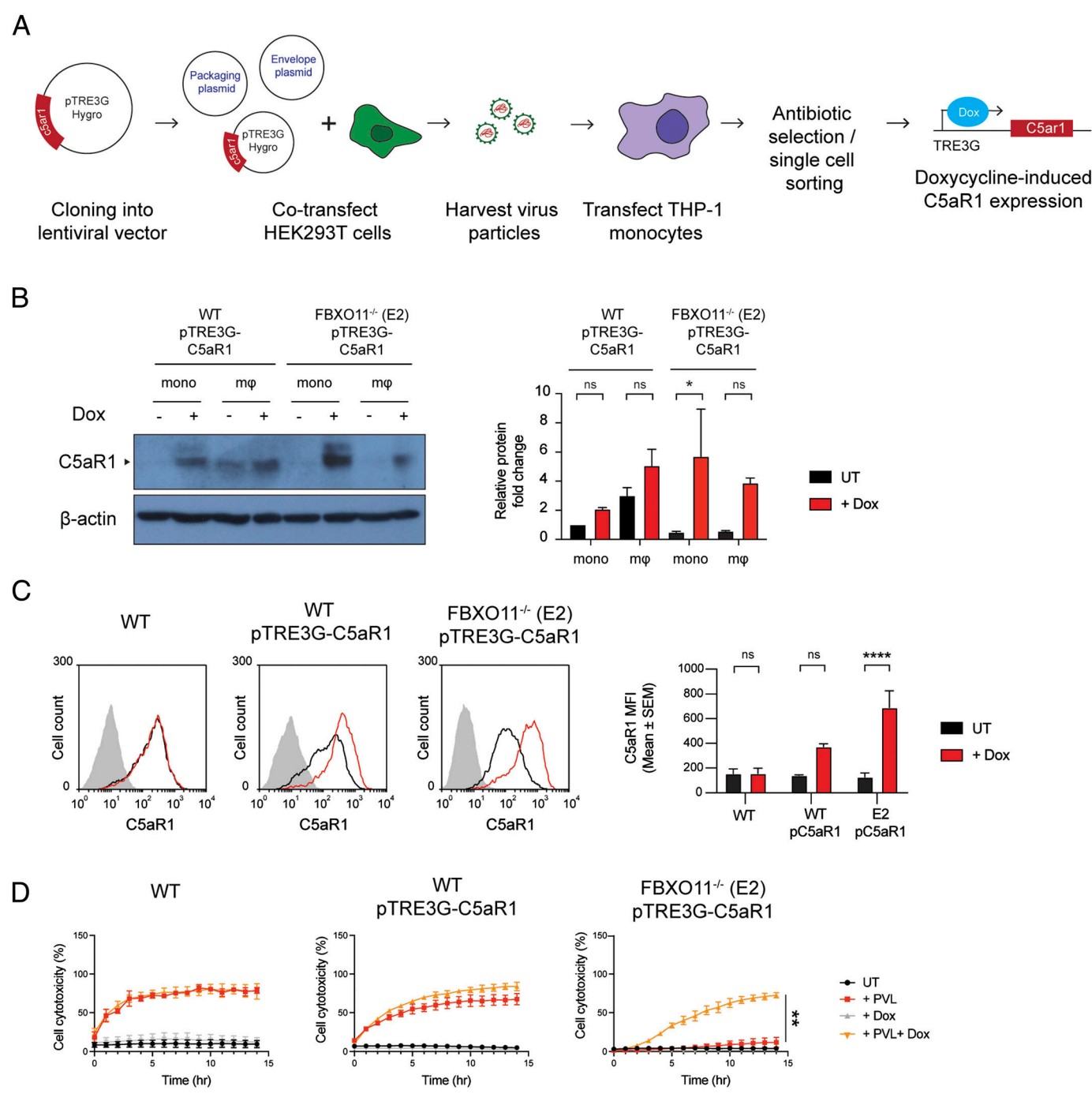

**Figure 3. Ectopic expression of C5aR1 restores cell susceptibility to PVL.**
**(A)** Graphical summary of the generation of doxycycline inducible C5aR1 overexpression THP1 monocytes using the lentiviral vector pTRE3G-C5aR1. **(B)** Western blot analysis of C5aR1 in pTRE3G-C5aR1 transfected cells—WT-pTRE3G-C5aR1 and FBXO11$^{-/-}$ (E2) pTRE3G-C5aR1 THP1 monocytes and macrophages. Cells were treated with or without doxycycline (1 μg/ml) for 24 h to induce C5aR1 expression. Abbreviations: macrophages (mΦ); monocytes (mono); doxycycline (Dox). Protein abundance was normalised to β-actin and represented as fold change compared with untreated cells. Mean ± SEM of three independent biological replicates shown. ns = not significant, * = $P < 0.05$; by two-way ANOVA with Sidak's multiple comparisons test. **(C)** Flow cytometric analysis and MFI of C5aR1 cell surface expression on WT and FBXO11$^{-/-}$ (E2) macrophages treated with (black line) or without (red line) doxycycline (1 μg/ml). The grey line represents isotype negative control. Representative of three independent experiments. ns = not significant, **** = $P < 0.0001$; by two-way ANOVA with Sidak's multiple comparisons test. **(D)** Live cell imaging showing the percentage of Draq7-positive (dead) untransfected WT macrophages, and pTRE3G-C5aR1 transfected WT and FBXO11$^{-/-}$ (E2) macrophages to PVL (125 ng/ml). Cells were treated with or without doxycycline (1 μg/ml) for 24 h before toxin treatment. Mean ± SEM of three independent experiments shown. ** = $P < 0.01$ for "+PVL" versus "+PVL +Dox" at 15 h post toxin treatment; by unpaired $t$ test.
Source data are available for this figure.

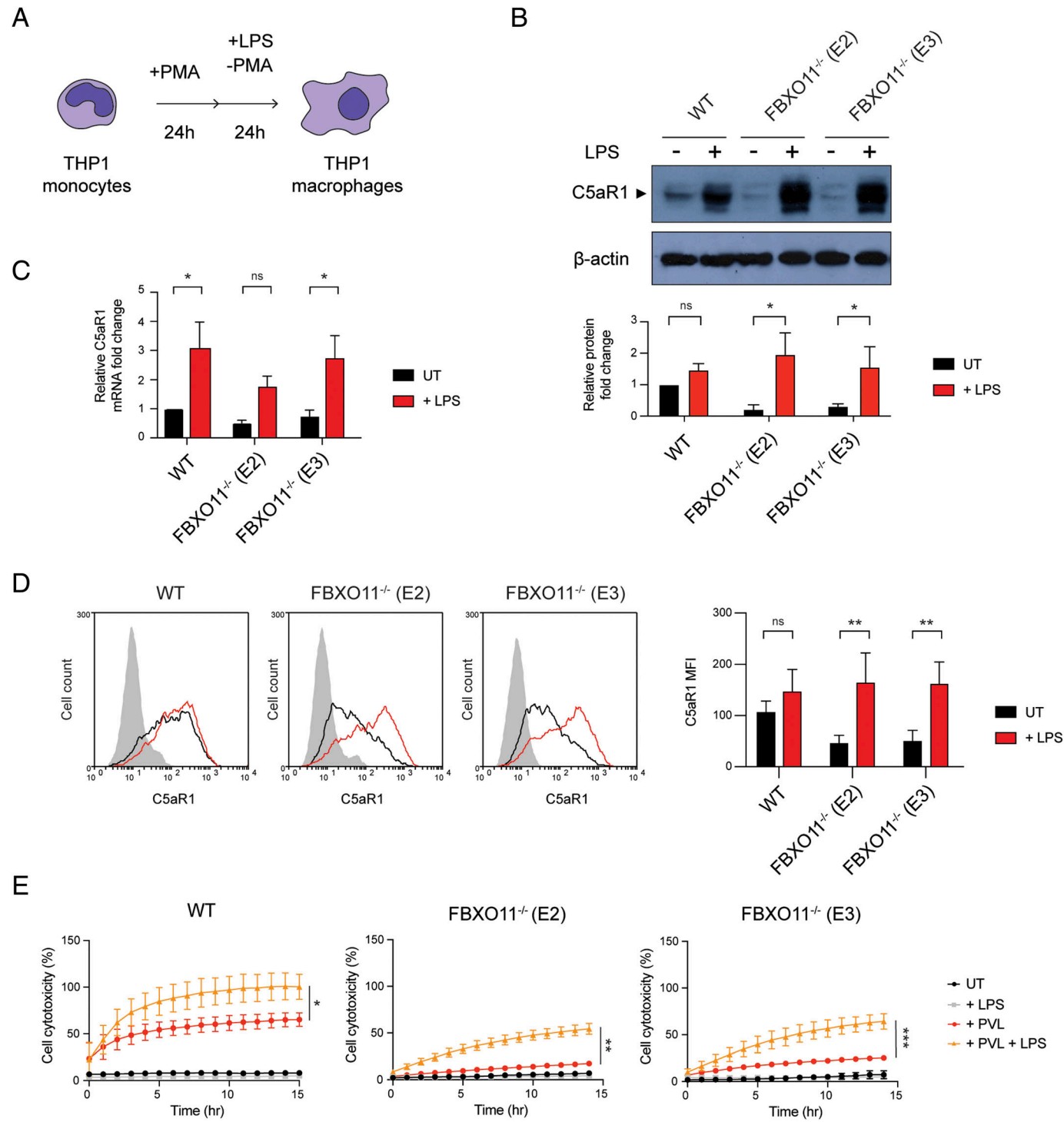

**Figure 4. LPS affects C5aR1 expression and PVL susceptibility in FBXO11-deficient macrophages.**
**(A)** Schematic illustration of THP1 PMA–induced differentiation and LPS treatment. **(B)** Western blot analysis of C5aR1 in WT and FBXO11$^{-/-}$ macrophages, treated with or without LPS for 24 h. Protein abundance was normalised to β-actin and represented as fold change compared with untreated WT macrophages. Mean ± SEM of three independent biological replicates shown. ns = not significant; * = $P < 0.05$; by two-way ANOVA with Sidak's multiple comparisons test. **(C)** qRT-PCR analysis of relative C5aR1 mRNA level, comparing the LPS-treated and -untreated macrophages. The mRNA levels were normalised to the control values of GAPDH, and fold change was LPS-treated cells that were compared with untreated WT macrophages. Mean ± SEM of three independent biological replicates shown. ns = not significant; * = $P < 0.05$; by two-way ANOVA with Sidak's multiple comparisons test. **(D)** Flow cytometric analysis and MFI of C5aR1 in LPS-treated or -untreated WT and FBXO11$^{-/-}$ macrophages. Grey line represents isotype, black line represents untreated, and red line represents LPS-treated macrophages. Mean ± SEM of three independent biological replicates shown. ns = not significant; ** = $P < 0.01$; by two-way ANOVA with Sidak's multiple comparisons test. **(E)** Live cell imaging showing the percentage of Draq7-positive (dead) LPS-

increased BCL-6 levels (Duan et al, 2012). Loss of FBXO11 also drives myeloid malignancies, but the underlying mechanism remains less well established (Yang et al, 2015; Mo et al, 2019). Our data show that the transcriptional regulation of *C5aR1* is independent of BCL-6. Ectopic expression or LPS-induced expression of C5aR1 in FBXO11$^{-/-}$ macrophages further demonstrates that C5aR1 is unlikely to be directly affected by FBXO11, despite the notion that post-translational modifications affect its protein activity and PVL cytotoxicity (Tromp et al, 2020). Analogous to the cell surface receptor CD40 in B cells, FBXO11 most likely controls *C5aR1* transcription, and future studies are aimed at identifying the other substrates besides BCL-6 of FBXO11 that control C5aR1 expression (Jiang et al, 2019).

Besides killing macrophages, PVL also activates the NLRP3 inflammasome, resulting in IL-1β secretion, an important pro-inflammatory cytokine involved in a wide range of autoimmune and inflammatory diseases. Our data demonstrate that FBXO11 regulates IL-1β secretion in response to bacterial toxins. Because of its central roles in immunity, IL-1β activity is regulated via several mechanisms. This includes the synthesis of an IL-1β inactive pro-form which is cleaved by caspase-1 after NLRP3 activation to produce the biologically active form and to enable IL-1β release via gasdermin D pores (Chan & Schroder, 2020). Given that nigericin kills WT and FBXO11-deficient THP1 macrophages with similar kinetics, this suggests that the NLRP3 inflammasome and subsequent membrane pore formation by gasdermin D are not directly regulated by FBXO11. Instead, FBXO11 dampened the expression of pro-IL-1β in naïve THP1 macrophages at the mRNA level. BCL-6 is a proto-oncogene with broad control over transcription of genes involved in not only B-cell lymphomagenesis but also cell cycle control, apoptosis, and inflammation (Kumagai et al, 1999; Saito et al, 2009; Basso & Dalla-Favera, 2010). BCL-6 negatively regulates NF-κB signalling, with BCL-6–deficient mice displaying dysregulated expression of cytokines including IL-6 and IL-1β (Li et al, 2005, 2020; Chen et al, 2017). Here, we further show that BCL-6 acts as a repressor for pro-IL-1β mRNA and protein levels but that other FBXO11-dependent mechanisms are at play.

More recently, we and others have identified that pro-IL-1β itself is polyubiquitinated and degraded by the proteasome, which allows for rapid reduction in the amount of IL-1β precursor available to be activated (Ainscough et al, 2014; Vijayaraj et al, 2021). However, pro-IL-1β was still turned over and ubiquitinated in the absence of FBXO11, suggesting that other E3 ligases are primarily involved in targeting IL-1β for degradation (Eldridge et al, 2017; Humphries et al, 2018). Regardless, our data show that loss of FBXO11 can result in increased inflammation downstream of inflammasome signalling due to increased pro-IL-1β levels. FBXO11 mutations in mice and humans are associated with ear inflammation due to otitis media (Hardisty-Hughes et al, 2006; Rye et al, 2011). Thus, it will be interesting to determine whether increased levels of pro-Il-1β underpin FBXO11-mediated inflammation in vivo.

In conclusion, our results reveal that FBXO11 transcriptionally regulates the expression of *C5aR1*, and that its absence can also

lead to dysregulated IL-1β expression. Further studies are required to identify the transcriptional factors involved in C5aR1 expression, and whether deletion of FBXO11 alters susceptibility to *S. aureus* infections. These findings demonstrate the utility of CRISPR screen in discovering and improving our understanding of the host–pathogen interactions. We provide rationale for further evaluation of FBXO11 as potential host-directed therapeutic target in infectious and inflammatory diseases.

# Materials and Methods

## Cell culture

Cas9-expressing THP1 (THP1-Cas9) cells were grown in RPMI 1640 media with 10% heat-inactivated fetal bovine serum (Bovogen), 25 mM HEPES (Sigma-Aldrich), and 1× penicillin–streptomycin (Gibco) at 37°C in 5% $CO_2$ in a humidified incubator. The cells were differentiated into macrophages using culture media supplemented with 160 nM of phorbol 12-myristate 13-acetate (PMA, Sigma-Aldrich) for 24 h. The media was replaced with fresh RPMI 1640 media without PMA, and the cells were allowed to recover for 24 h before treatments. HEK293T cells were grown in DMEM medium supplemented with 10% FBS, 1× GlutaMAX, and 1× pen/strep at 37°C in 5% $CO_2$. Cell lines were routinely verified to be mycoplasma negative by PCR.

## Generation of genome-wide mutant libraries and screening

The protocol was adapted from Yeung et al (2019). Briefly, a total of $2.4 \times 10^7$ THP1–Cas9 cells were transduced with the GeCKO v2.0 half library A at MOI of 0.3 and selected in puromycin for 14 d. Two independent pools of mutant library PMA-differentiated THP1–Cas9 macrophages were treated with PVL at 2 μg/ml for 24 h. Viable macrophages were collected by flow cytometry cell sorter and extracted for genomic DNA. PCR was used to amplify the gRNA region. Sequencing adaptors and barcodes were attached to the samples according to the conditions described by both Shalem et al and Sanjana et al, with some modifications (Sanjana et al, 2014). The samples were sequenced on a HiSeq 2500 (Illumina) to obtain 50-bp single end reads. In-house script was used to count the number of reads for each guide, and enrichment of guides and genes were analysed using the MaGeCK statistical package version 0.5.2 (Li et al, 2014). All gRNAs with a false-discovery rate of >0.0001 were removed, and combined candidate gene list based on genes with at least two gRNA hits in two experiments were generated.

## Monitoring macrophage death modalities using live cell imaging

THP1 monocytes were seeded at a density of $5 \times 10^5$ cells/ml and differentiated on 96-well tissue culture grade plate. For staining of

---

treated or -untreated WT and FBXO11$^{-/-}$ macrophages treated with PVL (62.5 ng/ml). Mean ± SEM of three independent biological replicates shown. * = *P* < 0.05, ** = *P* < 0.01, *** = *P* < 0.001 for + PVL versus + PVL + LPS at 14 h post toxin treatment; by unpaired *t* test.
Source data are available for this figure.

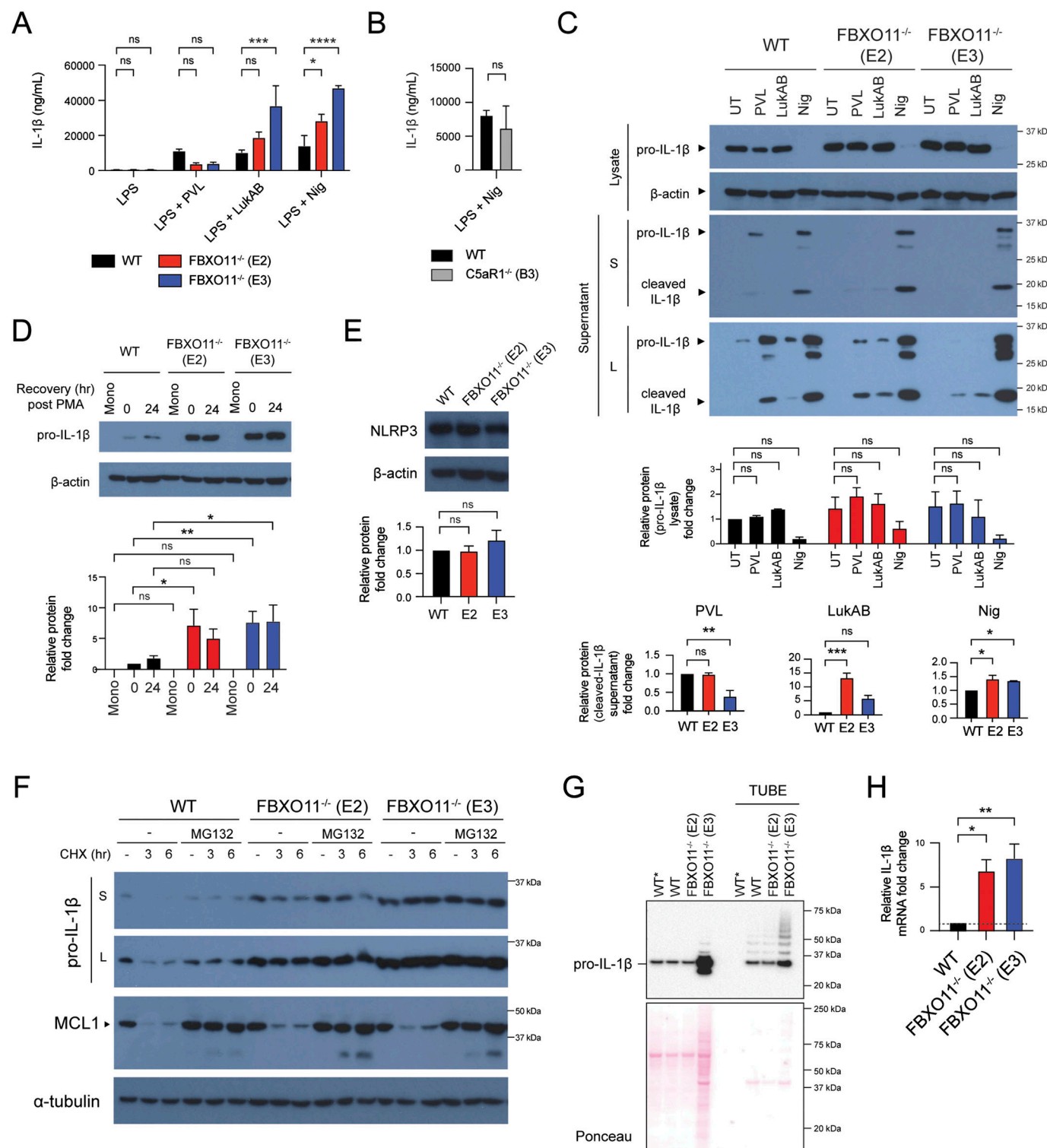

**Figure 5. FBXO11$^{-/-}$ macrophages exhibit higher IL-1β production.**
**(A, B)** WT, FBXO11$^{-/-}$, and (B) C5aR1$^{-/-}$ macrophages were primed with LPS (100 ng/ml) for 3 h before PVL (62.5 ng/ml), LukAB (15.6 ng/ml), or nigericin (10 µM) treatment for 2 h. Il-1β in culture supernatants were determined by ELISA. Mean ± SEM of three independent biological replicates shown. ns = not significant,; * = P < 0.05, *** = P < 0.001, **** = P < 0.0001; by two-way ANOVA followed by Sidak's multiple comparisons test. Abbreviation: Nig, nigericin. **(C)** Western blot analysis of pro-IL-1β (34 kD) and cleaved IL-1β (17 kD) in cell lysates and supernatants of WT and FBXO11$^{-/-}$ macrophages treated with LPS and/or toxins. Protein abundance was normalised to β-actin and represented as fold change compared with WT macrophages. Abbreviation: S, short exposure; L, long exposure. Mean ± SEM of three independent biological replicates shown. ns = not significant; by two-way ANOVA with Sidak's multiple comparisons test. **(D)** Western blot analysis of pro-IL-1β in whole cell lysates of unprimed THP1 cells. Monocyte (Mono) was differentiated with PMA, and macrophages were cultured in PMA-free media for the indicated amount of time. Protein abundance was

live cells, cells were treated with 1 mM of CellTracker Green (Thermo Fisher Scientific) for 30 min. The media was then replaced with RPMI 1640 media containing 600 nM Draq7 (Abcam) for 30 min. Cells were treated with indicated concentration of toxins, and/or reagents of choice and immediately imaged. Purified recombinant LukS-PV and LukF-PV were mixed in RPMI 1640 media for at least 5 min before addition to cells. Live cell imaging was performed using Leica DMi8 brightfield microscopy (Leica Microsystems) equipped with an incubator chamber set at 37°C in 5% $CO_2$. Time-lapse images were acquired with brightfield, GFP, and Y5 filters every 1 h overnight using 10×/0.32-NA objective lens. The percentage of Draq7-positive (dead) cells was quantified in ImageJ and MetaMorph (Molecular Devices) using custom-made journal suite incorporating the count nuclei function to segment and count the number of CTG- and Draq7-positive cells as described in the study by Speir et al, 2016 and Chow & Naderer, 2022. The MetaMorph journals are available at https://cloudstor.aarnet.edu.au/plus/s/8XGL750dIgwLYDn. The first frame of the GFP channel was used to calculate the total cell number. The percentage of Draq7-positive cells was analysed in Excel and GraphPad Prism 9.0.

### Quantitative real-time PCR (qRT-PCR)

THP1 monocytes were seeded at density of $1 \times 10^6$ cells/ml and differentiated on six-well tissue culture grade plates. The cells were lysed using TRIzol (Invitrogen) and treated with Turbo DNAse (Invitrogen) according to the manufacturer's instructions. RNA was reverse transcribed into cDNA using SuperScript III Reverse Transcriptase (Invitrogen) according to the manufacturer's instructions. qRT-PCR was performed on AriaMx Real-Time PCR system (Agilent) using FastStart Universal SYBR Green Master Rox reagent mix (Sigma-Aldrich) and analysed by relative quantification. The primer sets used for each gene are described in Table S2. Data were normalised to GAPDH, and ΔΔCq method was used to determine the fold change.

### Flow cytometry

THP1 monocytes ($1 \times 10^7$ cells) were seeded and differentiated on tissue culture grade Petri dish. The harvested cells were washed three times with PBS + 1% FBS and blocked on ice for 30 min. The cells were split into $10^6$ cells per 100 μl and incubated with human Fc block (BD Bioscience) at room temperature for 10 min. Cells were washed three times with PBS and incubated on ice for 30 min with primary antibodies—anti-C5aR1 (Lee et al, 2006), anti-C5L2 (Lee et al, 2006), anti-CD11b (#BMS104; Invitrogen), anti-CD45 (#304002;

BioLegend), or appropriate IgG isotype–matched controls. The cell pellets were washed three times and incubated with secondary antibodies, Alexa Fluor 488–conjugated goat anti-mouse or anti-rabbit IgG (#A28175; Thermo Fisher Scientific), on ice for 30 min. The cell pellets were washed three times, resuspended in 500 μl PBS + 1% FBS, and analysed on FACS Calibur (BD Bioscience). Gating was used to exclude cell debris and aggregates, and 10,000 events per sample were counted and visualized by histogram. The data were analysed using WEASEL flow cytometry software (Frank Battye).

To analyse LukS–PV binding, the harvested cells that were washed and blocked on ice were saturated with excess amount of LukS (2 μg/ml) on ice for 1 h after blocking step. The cell pellets were washed three times with PBS + 1% FBS and fixed using 4% (wt/vol) PFA for 10 min at room temperature and washed three times with PBS before incubation with anti-LukS-PV (in-house developed) for flow cytometric detection of LukS-PV on the cell surface using FACS Calibur.

### Immunofluorescence assay

THP1 monocytes were seeded at a density of $5 \times 10^5$ cells/ml and differentiated on sterile 12-mm thickness glass coverslips (#1217N79; Thermo Fisher Scientific). Cells were washed three times with PBS before fixation with 4% (wt/vol) paraformaldehyde for 10 min at room temperature. Cells were washed three times with PBS and treated with 50 mM $NH_4Cl$ for 10 min. After three washes in PBS, cells were incubated with anti-C5aR1 (clone 3C5) for 30 min at room temperature and washed three times with PBS. Cells were then incubated with blocking buffer (PBS + 3% bovine serum albumin) containing anti-mouse HRP secondary antibody and 0.1 μg/ml DAPI (Sigma-Aldrich) for 30 min. After three washes with PBS, coverslips were mounted on glass slides with fluorescence mounting medium (Dako) and dried at room temperature overnight. The slides were then sealed with nail polish and imaged on the Leica TCS SP5 confocal laser scanning microscope (Leica Microsystems). The images were analysed using ImageJ.

### Western blot analysis

THP1 monocytes were seeded at density of $1 \times 10^6$ cells/ml and differentiated on 24-well tissue culture grade plates. To harvest the whole-cell lysate, cells were first washed three times with PBS and 100 μl of 1× SDS-loading dye (10% [vol/vol] glycerol, 1% [wt/vol] SDS, 100 mM DTT, 0.005% [wt/vol] bromophenol blue, and 50 mM Tris–HCl [pH 6.8]) supplemented with a 1× protease inhibitor cocktail (Roche) were added. Samples were boiled for 5 min at

---

normalised to β-actin and represented as fold change compared with monocytes. Mean ± SEM of three independent biological replicates shown. ns = not significant; * = $P < 0.05$, ** = $P < 0.01$, *** = $P < 0.001$; by two-way ANOVA with Sidak's multiple comparisons test. **(E)** Western blot analysis of NLRP3 in whole-cell lysates of WT and FBXO11$^{-/-}$ macrophage. Protein abundance was normalised to β-actin and represented as fold change compared with WT macrophages. Mean ± SEM of three independent biological replicates shown. ns = not significant; by one-way ANOVA followed by Dunnett's multiple comparison test. **(F)** Western blot analysis of pro-IL-1β and MCL-1 in WT and FBXO11$^{-/-}$ macrophages. Cells were primed with LPS (100 ng/ml) for 2 h, with MG132 (20 μM) and Q-VD-OPh (20 μM) added in the last 30 min alongside. Cells were then treated with CHX (20 μg/ml) with or without MG132 for the indicated amount of time before cell lysate collection. Protein abundance was normalised to α-tubulin and represented as fold change compared with WT macrophages. Mean ± SEM of three independent biological replicates shown. ns = not significant; by two-way ANOVA with Sidak's multiple comparisons test. **(G)** Western blot analysis of pro-IL-1β in WT and FBXO11$^{-/-}$ macrophage cell lysates and TUBE-isolated ubiquitinated proteins. WT* indicates agarose beads only as control. **(H)** qRT-PCR analysis of relative IL-1β mRNA level. The mRNA levels were normalised to the control values of GAPDH. Mean ± SEM of three independent biological replicates shown. * = $P < 0.05$, ** = $P < 0.01$; by one-way ANOVA followed by Dunnett's multiple comparison test. Source data are available for this figure.

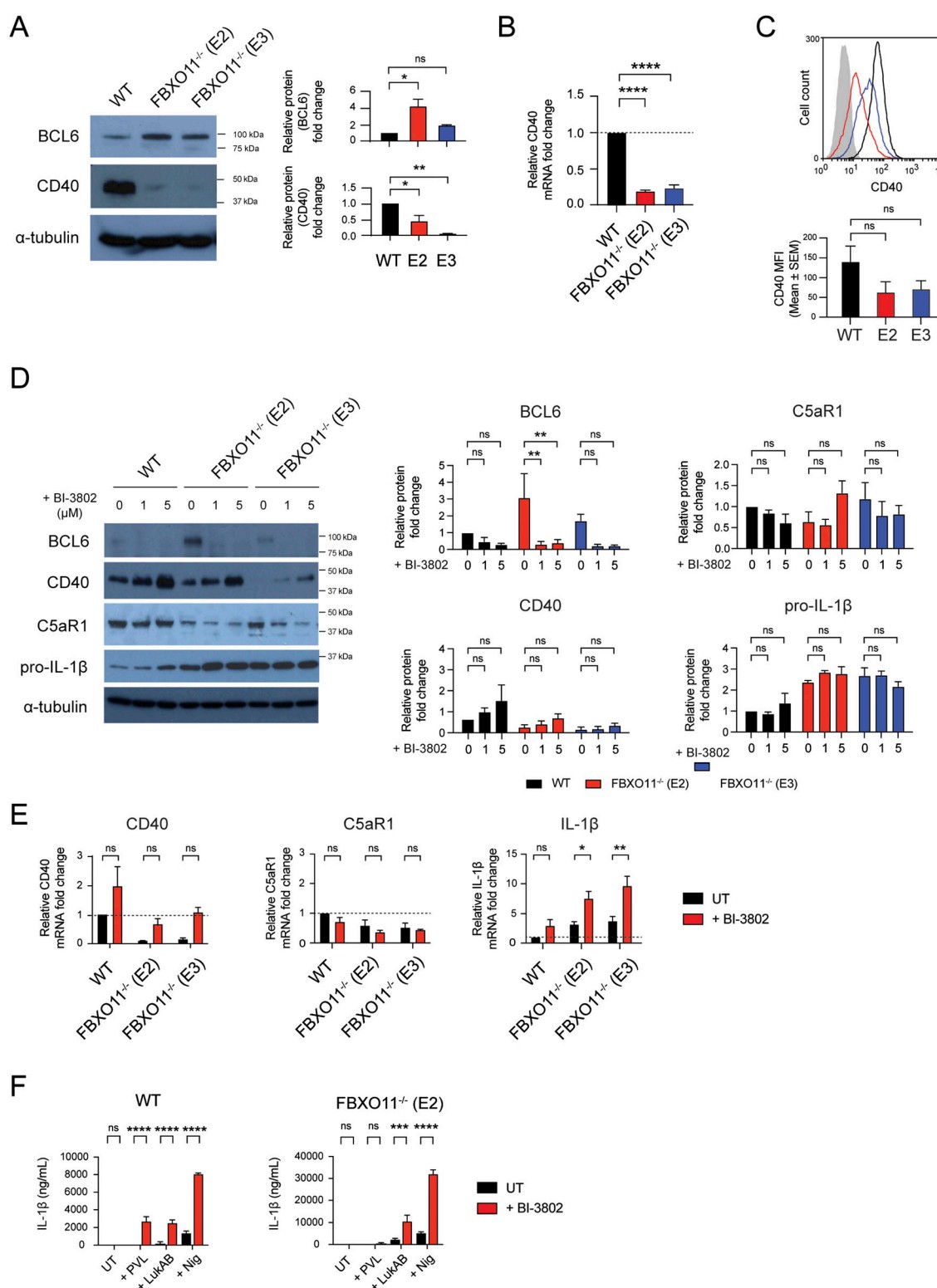

**Figure 6. Loss of BCL-6 promotes CD40 and IL-1β but not C5aR1 expression.**
**(A)** Western blot analysis of BCL-6 and CD40 in WT and FBXO11$^{-/-}$ macrophage whole-cell lysate. Protein abundance was normalised to α-tubulin and represented as fold change compared with untreated WT macrophages. Mean ± SEM of three independent biological replicates shown. ns = not significant, * = $P < 0.05$, ** = $P < 0.01$; by one-way ANOVA followed by Dunnett's multiple comparison test. **(B)** qRT-PCR analysis of CD40 mRNA in WT and FBXO11$^{-/-}$ macrophages. CD40 mRNA levels were normalised to GAPDH, and fold change relative to WT macrophage shown. Mean ± SEM of three independent biological replicates shown. **** = $P < 0.0001$; by one-way ANOVA followed by Dunnett's multiple comparison test. **(C)** Flow cytometric analysis and MFI of CD40 cell surface expression. Grey line represents isotype negative control, black line

95°C and separated on 10 or 12% SDS–PAGE gel. Proteins were transferred to nitrocellulose membranes (Bio-Rad) and blocked with 5% skim milk in TBST (0.2% Tween-20, 137 mM NaCl, 2.7 mM KCl, 25 mM Tris, pH 7.4) for 30 min. The membranes were then probed with primary antibodies — C5aR1 (#SC-271949; Santa-Cruz), LukS-PV, FBXO11 (# NB100-59826; Novus Biologicals), IL-1$\beta$ (#AF-401-NA; R&D Systems), BCL-6 (#PA-527390; Invitrogen), or CD40 (#ab224639; Abcam) — in 5% skim milk overnight at 4°C. Anti-$\beta$-actin (#8457; CST) or anti-$\alpha$-tubulin (#3873; CST) was used as a loading control. Membranes were washed three times with TBST and probed with goat anti-rabbit IgG (#R2004; Sigma-Aldrich), goat anti-mouse IgG (#M8642; Sigma-Aldrich), or rabbit anti-goat IgG (#A5420; Sigma-Aldrich) secondary antibody for 1 h at room temperature. After washing three times with TBST, membranes were developed using ECL reagent (Bio-Rad) before exposure to X-ray films (Kodak).

To analyse LukS-PV binding, PMA-differentiated THP1 macrophages were saturated with excess amount of LukS (2 $\mu$g/ml) suspended in RPMI 1640 media on ice for the indicated amount of time. The cell pellets were washed three times with PBS and 100 $\mu$l of 1× SDS-loading dye was added for whole-cell lysate collection.

### Lentiviral production and doxycycline induction

To construct the lentiviral vector, full-length human C5aR1 was amplified using high-fidelity DNA polymerase Phusion (NEB), as per the manufacturer's instructions. Primers used in this study are outlined in Table S2. Double digestion and ligation of the PCR product and the lentiviral vector pTRE3G-rtTAAD-Hygro containing a bidirectional doxycycline inducible promoter was performed with restriction enzymes BamHI-HF (NEB), NheI-HF (NEB), and T4 DNA ligase (NEB), as per the manufacturer's instructions. Plasmids were verified using DNA sequencing (primers outlined in Table S2) and transformed into *E. coli* DH5$\alpha$ competent cells. Plasmids were harvested for lentiviral vector production using GeneJET Plasmid Maxiprep Kit (Thermo Fisher Scientific), as per the manufacturer's instructions. The resultant plasmid containing the C5aR1 gene will be referred to as pTRE3G-C5aR1.

The lentivirus producer cell line HEK293T was transfected with pTRE3G-C5aR1 using Lipofectamine LTX (Invitrogen). In brief, cells were seeded on a six-well tissue culture plate the day before to obtain 70–90% confluency at the day of transfection. The media was replaced with Opti-MEM media (Gibco) 30 min before transfection. For each well, two solutions were prepared (12 $\mu$l of Lipofectamine LTX reagent diluted in 150 $\mu$l Opti-MEM, and 1.5 $\mu$g DNA, 1 $\mu$g packaging plasmid psPAX2, 0.5 $\mu$g envelope plasmid pMD2G, and 3 $\mu$l

Lipofectamine PLUS diluted in 150 $\mu$l Opti-MEM), which were mixed and incubated separately for 5 min at room temperature before being combined. The mixture was incubated for 30 min before being added dropwise to each well. The cells were incubated at 37°C with 5% $CO_2$ for 24 h before the media was replaced with fresh DMEM media without antibiotics. Viral supernatant was collected at 24 and 48 h post-infection and stored at –80°C. The efficacy of lentiviral transfection was analysed using expression of reporter eGFP gene as positive control.

THP1–Cas9 cells were seeded at a density of 2.5 × $10^5$ cells/ml in six-well tissue culture grade plates. The cells were transduced with the viral supernatants in the presence of polybrene (2 $\mu$g/ml) (Sigma-Aldrich) using flat fillet centrifuging (368$g$ for 1 h). After incubation of cells with viral supernatant for 24 h at 37°C with 5% $CO_2$, the media was replaced with fresh RPMI 1640 media and incubated for additional 24 h before being subjected to hygromycin selection (150 $\mu$g/ml) for 2 wk. After selection, cells were subjected to single cell sorting. For the induction of C5aR1 protein expression, cells were treated with doxycycline (1 $\mu$g/ml) for 24 h during recovery period.

### ELISA

THP1 monocytes were seeded at a density of 1 × $10^6$ cells/ml and differentiated on 96-well tissue culture grade plate. Cells were primed with or without LPS (100 ng/ml) for 3 h before treatment with toxins for 2 h. The cell culture supernatant was collected and centrifuged at 10,000 $g$ for 30 s. Human IL-1$\beta$/IL-1F2 DuoSet (R&D Systems) ELISA kit was used according to the manufacturer's instruction. The plate was measured for absorption at 450 nm with reference at 540 nm using a Tecan M200 plate reader (DKSH), and the cytokine concentration was analysed using Microsoft Excel and GraphPad Prism.

### TUBE pulldown assay

THP1 monocytes (1 × $10^7$ cells) were seeded and differentiated on tissue culture grade Petri dish and harvested. The cell pellets were lysed using DISC lysis buffer (30 mM Tris–HCl [pH 7.4], 120 mM NaCl, 2 mM EDTA, 2 mM KCl, 1% Triton X-100, Roche complete protease inhibitor cocktail, 2 mM NEM), and incubated on ice for 30 min. The cells were spun at top speed for 10 min, and the lysate was added to packed tube of TUBE or control agarose beads and incubated rotating overnight at 4°C. Beads with captured proteins were then washed with DISC buffer with protease inhibitor, resuspended in RSB buffer, and boiled at 85°C for 10 min for immunoblotting.

---

represents WT macrophages, and red and blue lines represent FBXO11$^{-/-}$ macrophages (E2 and E3, respectively). Mean ± SEM of three independent biological replicates shown. ns = not significant; by one-way ANOVA followed by Dunnett's multiple comparison test. **(D)** Western blot analysis of BCL-6, CD40, C5aR1, and IL-1$\beta$ in WT and FBXO11$^{-/-}$ macrophage whole-cell lysate treated with or without BI-3802 (1 or 5 $\mu$M) during recovery period. Protein abundance was normalised to $\alpha$-tubulin and represented as fold change compared with untreated WT macrophages at respective time points. Mean ± SEM of three independent biological replicates shown. ns = not significant; * = $P < 0.05$, ** = $P < 0.01$, *** = $P < 0.001$; by two-way ANOVA with Sidak's multiple comparisons test. **(E)** qRT-PCR analysis of CD40, C5aR1, and IL-1$\beta$ mRNA in WT and FBXO11$^{-/-}$ macrophages treated with or without BI-3802 (5 $\mu$M). mRNA levels were normalised to GAPDH, and fold change relative to WT macrophage shown. Mean ± SEM of three independent biological replicates shown. ns = not significant; * = $P < 0.05$, ** = $P < 0.01$; by two-way ANOVA with Sidak's multiple comparisons test. **(F)** BI-3802 (5 $\mu$M) untreated or treated WT, FBXO11$^{-/-}$ (E2) macrophages were treated with PVL (62.5 ng/ml), LukAB (62.5 ng/ml) or nigericin (10 $\mu$M) for 2 h. Il-1$\beta$ levels in culture supernatants were determined by ELISA. Mean ± SEM of three independent biological replicates shown. ns = not significant; ** = $P < 0.01$, *** = $P < 0.001$, **** = $P < 0.0001$; by two-way ANOVA followed by Sidak's multiple comparisons test. Abbreviation: Nig, nigericin.
Source data are available for this figure.

### In vitro *Staphylococcus* infection assays

*S. aureus* SF8300 (USA300) strains were grown in heart infusion (HI) broth at 37°C overnight, and then grown in fresh HI broth at 37°C to reach mid-log phase. The bacterial numbers before infection were determined in 1 ml PBS at $OD_{600nm}$, where an $OD_{600nm}$ of 1 corresponds to $1 \times 10^9$ bacteria per ml, which was used to determine the MOI.

PMA-differentiated macrophages were seeded at a density of $1.0 \times 10^6$ cells per mL into 24-well tissue culture plates. The macrophages were infected with heat-killed or live *S. aureus* SF8300 strains at an MOI of 10. After 30 min, the macrophages were washed three times in PBS, incubated in RPMI complete media supplemented with 10 µg/ml lysostaphin (Sigma-Aldrich) for 40 min. The macrophages were washed three times in PBS, and then incubated in RPMI complete media for 3 h. To harvest the whole-cell lysate, cells were first washed three times with PBS and 100 µl of 1× SDS-loading dye supplemented with a 1× protease inhibitor cocktail (Roche) were added. Samples were boiled for 5 min at 95°C and separated on 10% SDS–PAGE gel for Western blot analysis.

## Data Availability

All data are available in the main text, Supplementary Figures, and Tables. Source data are provided for relevant figures.

## Supplementary Information

## Acknowledgements

We acknowledge the expert help from members of Monash FlowCore and Micro Imaging facilities (Monash University). *S. aureus* wild-type strains SF8300 (USA300) was provided by Dr Binh Diep (University of California). The study was funded by the National Health and Medical Research Council (T Naderer, Grant number: 1163556) and the Australian Research Council (T Naderer, FT170100313). JE Vince was funded by an NHMRC Ideas grant (1183070) and Investigator grant (2008692).

### Author Contributions

Y Jeon: conceptualization, data curation, formal analysis, and investigation.

SH Chow: conceptualization, data curation, formal analysis, investigation, and writing—review and editing.

I Stuart: data curation.

A Weir: data curation.

ATY Yeung: conceptualization, data curation, and formal analysis.

C Hale: data curation and formal analysis.

S Sridhar: data curation.

G Dougan: formal analysis, supervision, and funding acquisition.

JE Vince: formal analysis, funding acquisition, investigation, and writing—review and editing.

T Naderer: conceptualization, formal analysis, supervision, funding acquisition, investigation, writing—original draft, and project administration.

### Conflict of Interest Statement

The authors declare that they have no conflict of interest.

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
