## [Reviewer comments · Life Science Alliance]

Life Science Alliance

FBXO11 governs macrophage cell death and inflammation in response to bacterial toxins

Yusun Jeon, Seong Chow, Isabella Stuart, Ashley Weir, Amy Yeung, Christine Hale, Sushmita Sridhar, Gordon Dougan, James Vince, and Thomas Naderer

DOI: <https://doi.org/10.26508/lsa.202201735>

Corresponding author(s): Thomas Naderer, Monash University

Review Timeline:	Submission Date:	2022-09-25
	Editorial Decision:	2022-11-04
	Revision Received:	2023-03-01
	Editorial Decision:	2023-03-15
	Revision Received:	2023-03-19
	Accepted:	2023-03-20

Scientific Editor: Novella Guidi

Transaction Report:

November 4, 2022

Re: Life Science Alliance manuscript #LSA-2022-01735-T

Thomas Naderer
Monash University
Department of Biochemistry & Molecular Biology
19 Innovation Walk
Clayton, VIC 3800

Dear Dr. Naderer,

Thank you for submitting your manuscript entitled "FBXO11 governs macrophage cell death and inflammation in response to bacterial toxins" to Life Science Alliance. The manuscript was assessed by expert reviewers, whose comments are appended to this letter. We invite you to submit a revised manuscript addressing the Reviewer comments.

Thank you for this interesting contribution to Life Science Alliance. We are looking forward to receiving your revised manuscript.

Sincerely,

B. MANUSCRIPT ORGANIZATION AND FORMATTING:

Reviewer #1 (Comments to the Authors (Required)):

In their manuscript, Jeon and colleagues identified a new role for the E3 ligase FBX011 in responses to bacterial toxins. They used a genetic approach to identify factors controlling the susceptibility to the Staph. Aureus toxin PVL. Their results suggest that FBX011 control cell surface expression of C5R, an important receptor mediating toxins effects. Lower C5R expression decrease cell death in response to PVL. Overall, the work is well performed and identify a new mechanism involving S.aureus pathogenesis.

The manuscript displays a large amount of work that help the authors make their points. Many interesting observations are also made, including the higher expression of IL1b by FBX011, suggesting that this E3 ligase may regulate its level at different level.

I have only a few elements to raise (and many question) from this very interesting piece of work.

Major comments:

1- I find the part on bcl6 a bit weaker as it still leaves it with an incomplete mechanism.
I believe the manuscript would be just fine without it.

Questions:

- 1- Are other isoforms of FBX011 expressed in your knockout?
- 2- Does FBX011 have been shown to transcriptionally control other proteins?
- 3- Is the E3 ligase activity of FBX011 required for the phenomenon observed?
- 4- Why is their more il-1b ubiquitinylation in FXB011 ko cells?
- 5- Does Bcl6 inhibition also increase secretion of IL-1b?
- 6- Does FBX011 also impact the expression level of other inflammasomes components (GSDMD, NLRP3, casp1, etc.)?

Reviewer #2 (Comments to the Authors (Required)):

The authors show elegantly with use of CRISPr technology that FBX011 influences killing of human macrophage cell lines by the Panton Valentine Leukocidin from Staphylococcus aureus. They also show that this is specific for PVL and not for other Staph toxins that do not target the C5aR1.
Their conclusion that the underlying mechanism is explained by the regulation of C5aR1 expression is clear and very well described.

The only thing that I find missing in the paper is how this all could relate to the pathophysiology of Staphylococcal infections. Does something during the infection trigger or downregulate FBX011? Is the entire pathway in some way activated or manipulated in the presence of the bacterium? Can we conclude that these events are related to infection by Staph for more in general in bacterial infections.

It is clear that the downstream effects are related to more (IL1beta) inflammatory events than PVL susceptibility alone, but the upstream events of FBX011 are unclear to me.

Additional experiments should address this issue and the consequences should be discussed in the discussion section.

Reviewer #3 (Comments to the Authors (Required)):

S aureus can cause severe infections like pneumonia and sepsis due to the presence of the pore-forming toxin Panton-Valentine

leukocidin (PVL) which causes leukocyte lysis, inflammation, and tissue necrosis. It is of interest to study this process as it may give rise to new druggable targets in the treatment of infectious and inflammatory diseases. Jeon et al have attempted to elucidate the mechanism of action of PVL toxicity by coming up with F-box protein 11 (FBXO11) as a possible master regulator of PVL actions. They have shown that the deleterious effects of PVL are not as severe in the absence of FBXO11. The authors have also shown that the effects of FBXO11 are a direct result of its regulation of the downstream targets C5aR1 and IL-1. To conclude the findings in this paper, reveal a potential therapeutic target for treatment of infectious and inflammatory diseases.

The premise of the work being done is here is potentially important however the manuscript in its current state does not do a good job of convincing the reader about the importance of FBXO11 as a novel target for potential new treatments. The rationale for selecting FBXO11 as a target is not sound enough. In addition, there have been quite a few conclusions made without providing adequate proof in the figures. Finally, the paper is somewhat haphazardly written with improperly framed sentences and multiple spelling mistakes. Specific suggestions are included below.

Suggestions

1. Figure 1B mentions the various cellular process the genes from the CRISPR screen were involved in. However, the reasons for selecting FBXO11 as a target have not been elucidated convincingly.
2. Line 250 mentions four sgRNAs however there are only 2 in Supplementary Table 1.
3. Lines 252 and 253 mention the isoform 4 encoding a 104kDa protein however the arrows in Fig 1d point to a protein closer to 150kDa. Also, the 94kDa isoform 1 band is barely visible in the Wild-type cells. Please mention the bands correctly or repeat the blots with a better antibody.
4. Line 268 = The gene counts are on the X-axis and not the Y-axis.
5. Figure 2B does not back up the author's claims of C5aR1 expression levels in FBXO11 ^{-/-} macrophages being similar to that of C5aR1 ^{-/-} macrophages. To make the claim of similar expression please show that the difference between them is not significant because just by looking at it there seems to be a significant difference between the two categories.
6. Supplementary Fig 1A shows a significant difference in CD 45 levels between the FBXO ^{-/-} clones compared to the WT. That contradicts the author's claims of the CD 45 levels being not or very mildly affected as mentioned in Lines 291, 292.
7. Supp Fig 1A shows an effect on CD 45 expression in FBXO11 ^{-/-} cells. So it would be interesting to see CD45 expression levels after C5aR1 expression levels were restored in Fig 3.
8. There seems to be a title missing for the section that begins on Line 384.
9. Please include the analysis of the transcription factor NF- κ B mentioned in Figure 4. That would be much more effective in making your point about the transcriptional regulation of C5aR1 in response to LPS treatment.
10. Please show the statistics for the LPS+PVL group in Fig 5a to make the claim in Lines 436, 437, 438. The authors also make the claim that LukAB and nigericin increased levels of mature IL-1 based on fig 5B. However, this is not true considering the complete or near absence of bands due to LukAb treatment in all the groups. It looks like LukAB increases the levels of the inactive pro-IL-1 while Nigericin as expected increases the active IL-1 levels.
11. Please show the quantification in all the western blots in the paper. In some blots the bands are barely visible while in some the bands look similar to each other. It would be much easier to understand the difference with the quantification.
12. Also, some bar graphs have statistical significance while some do not.
13. Finally, there are multiple grammatical mistakes in the paper including spelling mistakes (thrice spelled as trice) and improperly framed sentences like Line 555, 556. This makes the paper somewhat difficult to understand.

Our responses to each point raised by the reviewers are outlined in blue.

Reviewer #1 (Comments to the Authors (Required)):

In their manuscript, Jeon and colleagues identified a new role for the E3 ligase FBX011 in responses to bacterial toxins. They used a genetic approach to identify factors controlling the susceptibility to the Staph. Aureus toxin PVL. Their results suggest that FBX011 control cell surface expression of C5R, an important receptor mediating toxins effects. Lower C5R expression decrease cell death in response to PVL. Overall, the work is well performed and identify a new mechanism involving S.aureus pathogenesis.

The manuscript displays a large amount of work that help the authors make their points. Many interesting observations are also made, including the higher expression of IL1b by FBX011, suggesting that this E3 ligase may regulate its level at different level.

I have only a few elements to raise (and many question) from this very interesting piece of work.

Major comments:

1- I find the part on bcl6 a bit weaker as it still leaves it with an incomplete mechanism.

I believe the manuscript would be just fine without it.

To further strengthen our conclusions, we have performed additional experiments showing that BCL6 degradation results in increased IL-1 β expression and secretion in macrophages exposed to bacterial toxins (Fig 6F).

Questions:

1- Are other isoforms of FBX011 expressed in your knockout?

There are two known isoforms of FBX011 (NM_001190274 and NM_025133), both of which are not expressed in our knockout. As per reviewer #3's comment below, we have also included a longer exposure western blot (Fig 1E), as well as Fig S2A that better show the absence of the short isoform.

2- Does FBX011 have been shown to transcriptionally control other proteins?

In B-cells, FBX011 has been shown to indirectly control the transcription of proteins via targeted degradation of transcriptional factors such as BCL6 and CTBP1 (Jiang et al, Cell Reports, 2019). Future work is aimed at identifying the factors that control C5aR1 expression via FBX011 in macrophages.

3- Is the E3 ligase activity of FBX011 required for the phenomenon observed?

Although we have yet identified the direct FBXO11 target that leads to changes in C5aR1 expression, our data demonstrating BCL6 as a potential regulator of IL-1 β strongly suggests that the E3 ligase activity of FBXO11 is required for the observed phenomenon. In the future, we aim to perform ubiquitin ligase activity assays or mutagenesis study to test this further.

4- Why is there more IL-1 β ubiquitinylation in FBXO11 ko cells?

The higher intensity of pro-IL-1 β observed in the TUBE pulldown of FBXO11 (E3) is due to higher level of protein loading which did not affect our conclusion. Our hypothesis was that if FBXO11 was the E3 ligase of pro-IL-1 β , we should see a reduction/ablation in ubiquitinated proteins pulled down by the TUBE, which was not the case.

5- Does Bcl6 inhibition also increase secretion of IL-1 β ?

As mentioned above in our response to reviewer's major comment, we have performed an ELISA experiment (Fig 6F and Fig S3) and demonstrated that BCL6 inhibition does indeed lead to increased secretion of IL-1 β .

6- Does FBXO11 also impact the expression level of other inflammasomes components (GSDMD, NLRP3, casp1, etc.)?

We have demonstrated in Fig 5E, that there are no significant changes in NLRP3 protein level. Furthermore, nigericin kills FBXO11-deficient macrophages with similar kinetics as WT cells, suggesting that these cell death factors are not markedly affected (Fig 1G). We will examine expression levels of these factors in future studies.

Reviewer #2 (Comments to the Authors (Required)):

The authors show elegantly with use of CRISPR technology that FBXO11 influences killing of human macrophage cell lines by the Panton Valentine Leukocidin from *Staphylococcus aureus*. They also show that this is specific for PVL and not for other Staph toxins that do not target the C5aR1.

Their conclusion that the underlying mechanism is explained by the regulation of C5aR1 expression is clear and very well described.

The only thing that I find missing in the paper is how this all could relate to the pathophysiology of Staphylococcal infections. Does something during the infection trigger or downregulate FBXO11? Is the entire pathway in some way activated or manipulated in the presence of the bacterium? Can we conclude that these events are related to infection by Staph for more in general in bacterial infections.

It is clear that the downstream effects are related to more (IL1beta) inflammatory events than PVL susceptibility alone, but the upstream events of FBXO11 are unclear to me.

Additional experiments should address this issue and the consequences should be discussed in the discussion section.

We thank the reviewer for these suggestions. The primary aim of our CRISPR screen was to identify novel host factors specifically required for cell death by PVL toxin. Our screen does not reveal host factors that are directly triggered or downregulated during *S. aureus* infections, although we believe that some of the hits does have the potential to be hijacked by the bacterium.

While there are increasing evidence of exploitation of ubiquitin system by pathogens (i.e., SspH1 is a E3 ligase secreted by *Salmonella*, PMID: 34942035), there are currently no studies that have demonstrated whether infection processes up- or downregulate FBXO11 expression.

We have now performed additional experiments showing that protein levels of FBXO11 and C5aR1 in macrophages exposed to *S. aureus* exposure remains unaffected (Fig S2A).

We have discussed how FBXO11 could contribute to infections and inflammatory diseases. This is because FBXO11 is mutated in some individuals leading to cancers but also increased infections and inflammation.

Reviewer #3 (Comments to the Authors (Required)):

S. aureus can cause severe infections like pneumonia and sepsis due to the presence of the pore-forming toxin Pantone-Valentine leukocidin (PVL) which causes leukocyte lysis, inflammation, and tissue necrosis. It is of interest to study this process as it may give rise to new druggable targets in the treatment of infectious and inflammatory diseases. Jeon et al have attempted to elucidate the mechanism of action of PVL toxicity by coming up with F-box protein 11 (FBXO11) as a possible master regulator of PVL actions. They have shown that the deleterious effects of PVL are not as severe in the absence of FBXO11. The authors have also shown that the effects of FBXO11 are a direct result of its regulation of the downstream targets C5aR1 and IL-1 β . To conclude the findings in this paper, reveal a potential therapeutic target for treatment of infectious and inflammatory diseases.

The premise of the work being done is here is potentially important however the manuscript in its current state does not do a good job of convincing the reader about the importance of FBXO11 as a novel target for potential new treatments. The rationale for selecting FBXO11 as a target is not sound enough. In addition, there have been quite a few conclusions made without providing adequate proof in the figures. Finally, the paper is somewhat haphazardly written with improperly framed sentences and multiple spelling mistakes. Specific suggestions are included below.

Suggestions

1. Figure 1B mentions the various cellular process the genes from the CRISPR screen were involved in. However, the reasons for selecting FBXO11 as a target have not been elucidated convincingly.

Our reasoning for selecting FBXO11 for investigation was largely based on two factors: significant reduction in toxin susceptibility and targets previously unidentified in other similar screens. To emphasize our reasoning, we have revised the text and added citations to paper that have demonstrated the ability of bacteria to hijack host ubiquitination system to emphasise its importance in host-pathogen interaction.

2. Line 250 mentions four sgRNAs however there are only 2 in Supplementary Table 1.

We have revised the text.

3. Lines 252 and 253 mention the isoform 4 encoding a 104kDa protein however the arrows in Fig 1d point to a protein closer to 150kDa. Also, the 94kDa isoform 1 band is barely visible in the Wild-type cells. Please mention the bands correctly or repeat the blots with a better antibody.

While the predicted protein size of the isoforms is 104 and 94kDa, the antibody used routinely detect higher molecular sized proteins in our hands. This may reflect the cells used (macrophages versus B-cell or HeLa cells). We have addressed such discrepancy observed in the result section. We have also included a long-exposure blot that better shows the isoform 1 (Fig 1E).

4. Line 268 = The gene counts are on the X-axis and not the Y-axis.

We have corrected the error in the figure legend.

5. Figure 2B does not back up the author's claims of C5aR1 expression levels in FBXO11 ^{-/-} macrophages being similar to that of C5aR1 ^{-/-} macrophages. To make the claim of similar expression please show that the difference between them is not significant because just by looking at it there seems to be a significant difference between the two categories.

We have modified the wording to better reflect that "C5aR1 expression is 'markedly reduced in FBXO11^{-/-}, though not completely absent as in C5aR1^{-/-} cells".

6. Supplementary Fig 1A shows a significant difference in CD 45 levels between the FBXO ^{-/-} clones compared to the WT. That contradicts the author's claims of the CD 45 levels being not or very mildly affected as mentioned in Lines 291, 292.

We have reworded the sentence to state that "While levels of CD45 were also decreased in FBXO11^{-/-} macrophages, it remained readily detectable on the cell surface (Fig S1)."

7. Supp Fig 1A shows an effect on CD 45 expression in FBXO11 ^{-/-} cells. So it would be interesting to see CD45 expression levels after C5aR1 expression levels were restored in Fig 3.

We have now included additional experimental data (Fig S2B) investigating the CD45 expression after C5aR1 expression was restored and have found no significant changes.

8. There seems to be a title missing for the section that begins on Line 384.

We have now included an additional title "LPS increases C5aR1 expression and restores susceptibility of FBXO11^{-/-} macrophages to PVL-mediated death".

9. Please include the analysis of the transcription factor NF- κ B mentioned in Figure 4. That would be much more effective in making your point about the transcriptional regulation of C5aR1 in response to LPS treatment.

The study by Hunt *et al* (2005) identified the putative CCAAT site within the promoter region of C5aR1, which putatively binds the transcriptional activator NF- κ B. The aim of our experiments was to simply test whether the loss of FBXO11 affects the promoter activity of C5aR1 gene, and whether C5aR1 level can still be restored in the KO cells. Further investigation of the transcriptional regulation of C5aR1 is of interest, but we feel that this is beyond the scope of the current manuscript.

10. Please show the statistics for the LPS+PVL group in Fig 5a to make the claim in Lines 436, 437, 438. The authors also make the claim that LukAB and nigericin increased levels of mature IL-1 β based on fig 5B. However, this is not true considering the complete or near absence of bands due to LukAb treatment in all the

groups. It looks like LukAB increases the levels of the inactive pro- IL-1 β while Nigericin as expected increases the active IL-1 β levels.

We have now included a longer exposure of the immune blot to more clearly show that LukAB causes mature Il1beta secretion, which is futher increased in FBXO11 deficient macrophages. We have now included quantification and p-values, and reworded the text to clearly indicate this.

11. Please show the quantification in all the western blots in the paper. In some blots the bands are barely visible while in some the bands look similar to each other. It would be much easier to understand the difference with the quantification.

We have now included longer exposures of western blots and quantified bands from at least three independent experiments, which is included next to the western blots.

12. Also, some bar graphs have statistical significance while some do not.

We have included non-significant (ns) to reflect that remaining samples do not differ significantly.

13. Finally, there are multiple grammatical mistakes in the paper including spelling mistakes (thrice spelled as trice) and improperly framed sentences like Line 555, 556. This makes the paper somewhat difficult to understand.

We have corrected the grammatical mistakes.

March 15, 2023

RE: Life Science Alliance Manuscript #LSA-2022-01735-TR

Dr. Thomas Naderer
Monash University
Department of Biochemistry & Molecular Biology
19 Innovation Walk
Clayton, VIC 3800
Australia

Dear Dr. Naderer,

Thank you for submitting your revised manuscript entitled "FBXO11 governs macrophage cell death and inflammation in response to bacterial toxins". We would be happy to publish your paper in Life Science Alliance pending final revisions necessary to meet our formatting guidelines.

- please upload your main manuscript text as an editable doc file
- please use the [10 author names, et al.] format in your references (i.e. limit the author names to the first 10)
- please add a separate figure legend section to your main manuscript
- please add a figure callout for Figure 4A to your main manuscript text

A. FINAL FILES:

B. MANUSCRIPT ORGANIZATION AND FORMATTING:

Sincerely,

Reviewer #1 (Comments to the Authors (Required)):

The authors have addressed my concerns. I believe this story is strong and will be of great interest to the scientific community.

Reviewer #2 (Comments to the Authors (Required)):

The manuscript has now been improved, my questions were answered, I have no further questions or remarks

Reviewer #3 (Comments to the Authors (Required)):

Having thoroughly read the author's responses as well as the manuscript I will say that my concerns have definitely been addressed.

I have no further questions or comments at this time.

March 20, 2023

RE: Life Science Alliance Manuscript #LSA-2022-01735-TRR

Dr. Thomas Naderer
Monash University
Department of Biochemistry & Molecular Biology
19 Innovation Walk
Clayton, VIC 3800
Australia

Dear Dr. Naderer,

Thank you for submitting your Research Article entitled "FBXO11 governs macrophage cell death and inflammation in response to bacterial toxins". It is a pleasure to let you know that your manuscript is now accepted for publication in Life Science Alliance. Congratulations on this interesting work.

DISTRIBUTION OF MATERIALS:

Again, congratulations on a very nice paper. I hope you found the review process to be constructive and are pleased with how the manuscript was handled editorially. We look forward to future exciting submissions from your lab.

Sincerely,
